# Assessment of Preventative Measures Practice among Umrah Pilgrims in Saudi Arabia, 1440H-2019

**DOI:** 10.3390/ijerph18010257

**Published:** 2020-12-31

**Authors:** Mansour Tobaiqy, Ahmed H. Alhasan, Manal M. Shams, Samar A. Amer, Katie MacLure, Mohammed F. Alcattan, Sami S. Almudarra

**Affiliations:** 1Department of Pharmacology, College of Medicine, University of Jeddah, Jeddah 21512, Saudi Arabia; 2Jeddah Eye Hospital, Ministry of Health, Jeddah 23331, Saudi Arabia; Ahhalhasan@moh.gov.sa; 3Health and Lifestyle Department, Ministry of Health (MOH), Riyadh 11176, Saudi Arabia; mmahmoud-shams@moh.gov.sa; 4Public Health and Community Medicine, Zagazig University, Zagazig 44519, Egypt; amers@moh.gov.sa; 5Public Health, Ministry of Health (MOH), Riyadh 11176, Saudi Arabia; 6Independent Research Consultant, Aberdeen AB32 6RU, UK; katiemaclure@outlook.com; 7Ministry of Health (MOH), Jeddah 23222, Saudi Arabia; malcattan@moh.gov.sa; 8Epidemiology, Surveillance and Preparedness, Saudi CDC, General Supervisor of Saudi Field Epidemiology Training Program, Ministry of Health (MOH), Riyadh 11176, Saudi Arabia; salmudarra@moh.gov.sa

**Keywords:** Umrah, pilgrims, personal preventative measures, COVID-19, Makkah, Kingdom of Saudi Arabia

## Abstract

**Background:** Annually, approximately 10 million pilgrims travel to the Kingdom of Saudi Arabia (KSA) for Umrah from more than 180 countries. This event presents major challenges for the Kingdom’s public health sector, which strives to decrease the burden of infectious diseases and to adequately control their spread both in KSA and pilgrims home nations. The aims of the study were to assess preventative measures practice, including vaccination history and health education, among Umrah pilgrims in Saudi Arabia. **Methods:** A cross sectional survey was administered to pilgrims from February to April 2019 at the departure lounge at King Abdul Aziz International airport, Jeddah city. The questionnaire comprised questions on sociodemographic information (age, gender, marital status, level of education, history of vaccinations and chronic illnesses), whether the pilgrim had received any health education and orientation prior to coming to Saudi Arabia or on their arrival, and their experiences with preventative practices. **Results:** Pilgrims (*n* = 1012) of 41 nationalities completed the survey. Chronic diseases were reported among pilgrims (*n* = 387, 38.2%) with cardiovascular diseases being the most reported morbidity (*n* = 164, 42.3%). The majority of pilgrims had been immunized prior to travel to Saudi Arabia (*n* = 770, 76%). The most commonly reported immunizations were influenza (*n* = 514, 51%), meningitis (*n* = 418, 41%), and Hepatitis B virus vaccinations (*n* = 310, 31%). However, 242 (24%) had not received any vaccinations prior to travel, including meningitis vaccine and poliomyelitis vaccine, which are mandatory by Saudi Arabian health authorities for pilgrims coming from polio active countries. Nearly a third of pilgrims (*n* = 305; 30.1%) never wore a face mask in crowded areas during Umrah in 2019. In contrast, similar numbers said they always wore a face mask (*n* = 351, 34.6%) in crowded areas, while 63.2% reported lack of availability of face masks during Umrah. The majority of participants had received some form of health education on preventative measures, including hygiene aspects (*n* = 799, 78.9%), mostly in their home countries (*n* = 450, 44.4%). A positive association was found between receiving health education and practicing of preventative measures, such as wearing face masks in crowded areas (*p* = 0.04), and other health practice scores (*p* = 0.02). **Conclusion:** Although the experiences of the preventative measures among pilgrims in terms of health education, vaccinations, and hygienic practices were at times positive, this study identified several issues. These included the following preventative measures: immunizations, particularly meningitis and poliomyelitis vaccine, and using face masks in crowded areas. The recent COVID-19 pandemic highlights the need for further studies that focus on development of accessible health education in a form that engages pilgrims to promote comprehensive preventative measures during Umrah and Hajj and other religious pilgrimages.

## 1. Introduction

Around 10 million pilgrims travel annually to the Kingdom of Saudi Arabia (KSA) for Umrah (a minor pilgrimage to Makkah) from more than 180 countries [1]. According to The Ministry of Hajj and Umrah, the government of KSA has issued 7,584,424 Umrah visas since 30 May 2019 (25 Ramadan 1440 H), with pilgrims coming via air, land, or sea to visit the holy cities of Makkah and Madinah [1,2].

Umrah is a spiritual practice that is not obligatory and has no fixed date or time to be performed so may be seasonally affected [3]. Similarly to the Hajj and other mass gatherings, the Umrah entails global movement of people, gathering of crowds, and therefore an increased health challenge, including the risk of spreading infection [4]. Respiratory tract infections, overcrowding accidents, pollution, and many other medical and health issues are encountered by the host country, which requires special attention and measurable responses to mitigate these risks [4,5]. Without proper planning and preventative measures, this religious activity can overwhelm the health system of the host country and impact global health preparedness, because the majority of Umrah pilgrims come from, and return to, their different countries [5,6].

The Umrah performer (pilgrim), called hajji in the Arabic language, must be received and accompanied by a company authorized by the Ministry of Hajj, called “Al Hamla”, which provides Hajj and Umrah services including catering, transportation, travel arrangements, accommodation, and first aid health services for every pilgrim who has paid for their services [1,6].

Before travelling to Saudi Arabia for Umrah, pilgrims are advised to seek medical advice about potential health risks and bring vaccination certificates to be inspected at the entry port by Saudi authorities [3]. Even before the COVID-19 pandemic, the Saudi Arabian Ministry of Health (MOH) published a guide for pilgrims on health issues and standard vaccination requirements, together with important control measures for the safety of the individuals and prevention of infectious disease outbreaks [7].

There is no doubt that providing health education to pilgrims on non-communicable and infectious diseases (including the preventative measures and modes of infection transmission) is of great interest for the Hajj and health authorities in Saudi Arabia [5]. It is also critical for global public health and disease control, which is considered good practice to minimize the risks and improve compliance with preventative control measures [8,9,10].

During Umrah, pilgrims have a vital role in sustaining public health through practicing preventative measures such as personal hygiene, catching coughs and sneezes, hand washing with water and soap or sanitizers, wearing face masks (and doing so properly), and subsequent waste disposal of materials in sanitary bins [5,7]. Face mask use was considered an affordable and effective method to control the exposure to pathogens in high-risk environments, to reduce the risk of transmittable infectious diseases including COVID-19 [11], and to protect from the inhalation of aerosols containing organic and inorganic particulates [12]. More recent studies cast doubt on the effectiveness of masks for preventing infection transmission, with some studies reporting over 90% of pilgrims experiencing respiratory tract infections (RTI), suggesting preventative measures are needed [13,14,15,16,17]. Hoang et al. (2019) conducted sequential systematic testing of 121 French pilgrims during Hajj 2018 [18]. Amongst their sample, 40.5% reported wearing a face mask, 55.4% washing hands more often, and 87.6% using disposable tissues. They found RTIs at the Hajj were as “a result of complex interactions between a number of respiratory viruses and bacteria” with 93.4% (*n* = 113/121) experiencing at least one respiratory infection despite preventative measures [18]. The same research team conducted a smaller study focused on COVID-19 at the 2020 Hajj, which had restricted attendance (*n* = 1000) with only Saudi nationals permitted [19]. The wearing of face masks was mandated with no public health emergency noted when the Hajj ended early August 2020 [19]. However, a recent systematic review of 21 studies by Liang et al. (2020) found “the risk of influenza, SARS, and COVID-19 infection were reduced by 45%, 74%, and 96% by wearing masks, respectively” [20]. The evidence is both contradictory and inconclusive [13,14,15,16,17,18,19,20].

On 27 February 2020, the KSA Foreign Ministry took a decisive and proactive preventative measure. To prevent the spread of COVID-19, it suspended Umrah visas, denying entry of pilgrims to the holy cities and the entire Kingdom [2].

The Saudi Arabian Vision 2030 aimed to build outstanding health and economic services for visitors to the country, including Umrah and Hajj pilgrims. In light of an initiative called “To care about Rahman Guests” introduced by the Ministry of Health Community Services in Jeddah, in collaboration with a voluntary group (Vision Team) and in view of the current global pandemic of COVID-19 that has affected millions of people worldwide, there is an imperative need for effective preventative measures to be adopted nationally to increase preparedness for future Umrah and Hajj.

Given the clear importance of controlling public health and infectious diseases at mass religious gatherings, underlined by the COVID-19 pandemic, this study aimed to assess the experiences of preventative measures practiced among Umrah pilgrims following their Hajj in Saudi Arabia.

## 2. Methods

### 2.1. Study Design

A cross-sectional survey was conducted among Umrah pilgrims before departure from February to the end of April 2019 at King Abdul Aziz Airport-Islamic port in Jeddah, Saudi Arabia.

### 2.2. Questionnaire Development

The questionnaire (Appendix A) was composed of sociodemographic items including age, gender, nationality, level of education, history of vaccinations and chronic illnesses; whether the pilgrim had received any health education and orientation prior to coming to Saudi Arabia; or upon their arrival, their experiences of the practice of preventative measures such as using umbrellas, hats, or sunscreen to protect from excessive heat and sun, use of hygiene tools and practices such as wearing of face masks and washing hands, and the availability of relevant resources. Pilgrims were also asked to rate their level of satisfaction around the health services provided by the Saudi Arabian MOH or other providers.

Item types included closed questions, where the tick list was originated from a combination of previous research on mass gatherings and public health issues, with respondents permitted to select more than one response in some questions.

The questionnaire was reviewed for face and content validity by members of the research team and experts in public health in the MOH. Validity testing was followed by piloting with twenty Umrah pilgrims of different nationalities and spoken languages, with no changes required post-piloting. Therefore, the pilot responses were included in the analysis dataset for reporting. The questionnaire was written in Arabic and was translated by the Al Hamla company’s supervisors to the pilgrim’s specific languages to be easily understood. Finally, all responses were gathered by the multi-lingual research team and translated to English for analysis and reporting.

### 2.3. Recruitment

Potential participants were approached opportunistically by a research team member from February until the end of April 2019 at the departure lounge at King Abdul Aziz International Airport in Jeddah city. They were asked if they had time to participate in a short survey concerning the preventative measures in Umrah practice. Those who agreed were administered the questionnaire by one of the 22 trained personnel who recorded the responses electronically into an automated dataset that excluded incomplete questionnaires. Participants had to be at least 18 years of age; there were no exclusion criteria.

### 2.4. Analysis

Statistical analysis was undertaken using SPSS (IBM SPSS Statistics for Windows Inc, Cart, NY, USA, version 22). The data were summarized and analyzed using descriptive and inferential tests. A *t*-test for independent samples was conducted to identify the relationship between receiving health education and the practice of preventative measures by pilgrims; *p* < 0.05 was considered statistically significant.

## 3. Results

1012 questionnaires were completed by the participants with 41 nationalities represented.

### 3.1. Sociodemographic Characteristics

As shown in Table 1, the majority of participants were male (*n* = 656, 64.8%), with 34.6% female (*n* = 356), and married (*n* = 773, 76.3%). Pilgrims were from Iraq (*n* = 177, 17.5%) followed by Egypt (*n* = 161, 15.9 %), Sudan (*n* = 113, 11.1%), Indonesia (*n* = 111, 11%) and Pakistan (*n* = 92, 9.1%), respectively, as per Table 2.

### 3.2. Level of Education

While the majority had attended high school or above (*n* = 622, 61.4%), there was a proportion with only a basic level of reading and writing ability (*n* = 94, 9.3%), with some describing themselves as illiterate (*n* = 58, 5.7%).

### 3.3. Medical Condition

Chronic disease including multimorbidity was reported among pilgrims. Cardiovascular diseases were the most prevalent (*n* = 164, 42.4%), followed by diabetes mellitus (*n* = 139, 37%), as per Table 1.

### 3.4. Immunization History

Despite the Umrah visa requirements, some pilgrims (*n* = 223; 22%) had not received any vaccination prior to travel; however, others had received influenza vaccine (*n* = 514; 50.8%), meningitis vaccination (*n* = 418; 41.3%), Hepatitis B vaccine (HBV) (*n* = 310; 30.6%), and poliomyelitis vaccination (*n* = 285; 28.1%) (Table 3).

In the case of poliomyelitis vaccination for pilgrims attending from countries listed at the time of the 2019 Umrah as polio active, some of the pilgrims had not taken a polio vaccine before travelling from at risk countries including Pakistan (*n* = 34/92; 31.2%) and Afghanistan (*n* = 16/80; 12.8%) [21].

### 3.5. Duration of Stay in Makkah and Madinah

The majority of participants stayed in Makkah for less than two weeks (*n* = 660, 65.2%) and in Madinah for less than a week (*n* = 614, 60.6%) (Table 3).

### 3.6. Health Education

The majority of pilgrims had received some form of health education (*n* = 799, 78.9%) described as self-education (*n* = 185, 23.2%); however, a proportion (*n* = 123, 12.1%) indicated that they had not received any (Table 4). The majority received health education in their homeland (*n* = 450, 44.4%) or on arrival in Saudi Arabia (*n*= 201, 19.8%). Many believed that the health education they had received was helpful (*n* = 661, 65.3%) or helpful “to some extent” (*n* = 216; 21.3%); however, some pilgrims (*n* = 135; 13.3%) did not find the health education helpful. The most cited source of health education was lectures (*n* = 262; 25.8%), followed by travel clinics (*n* = 134; 13.2%), followed by family and friends (*n* = 127, 12.5%), followed by health care providers (*n* = 122, 12%), as per Table 4.

### 3.7. Practice of Preventative Measures

Nearly a third of pilgrims (*n* = 305; 30.1%) never wore a face mask in crowded areas during Umrah in 2019. In contrast, similar numbers said they always wore a face mask (*n* = 351, 34.6%) in crowded areas. The majority of pilgrims (*n* = 840, 82.9%) did wash their hands with soap and water or sanitizers after coughing and sneezing. Most pilgrims reported that they never used an umbrella or hat to protect them from the sun (*n* = 509, 50.2%), while some always used an umbrella or hat (*n* = 144, 14.2%) (Table 5).

Table 6 shows the responses to items on aspects of availability and the sources of Personal Protective Equipment. Many pilgrims reported face masks were not available (*n* = 642, 63.4%).

Table 7 reports the association between receiving some form of health education and practicing of preventative measures by pilgrims. Receiving health education was significantly associated with the wearing of face masks in crowded areas (*p* = 0.04) and healthy practice score (i.e., washing hands with water, soap, or sanitizers, use of personal tools, disposing of used tissues in waste bin) (*p* = 0.02).

Table 8 indicates that the majority of pilgrims were very satisfied with the Saudi Arabian MOH Services (*n* = 734, 72.9%).

## 4. Discussion

One thousand and twelve pilgrim of 41 nationalities who performed Umrah in the Saudi Arabia in 2019 had completed the survey, almost a quarter of whom had not received any vaccination prior to travel to the holy city, including meningitis vaccine and poliomyelitis vaccine, which are mandatory by Saudi Arabian health authorities. Another issue is that a considerable percentage of pilgrims (30.1%) never wore a face mask in crowded areas during Umrah, which merits further investigations for the causes and improvement polices, particularly amid the COVID-19 pandemic.

Notably, all pilgrims arriving Saudi Arabia must have received a single dose of quadrivalent meningococcal vaccine and provide proof of a valid vaccination or prophylaxis in the five years before arrival to allow the issue of a Hajj or Umrah visa by the Saudi Arabian Authorities [7]. However, this study reported less than half of the pilgrims (*n* = 418, 41.3%) surveyed actually had this vaccination, an issue that has been raised before by Ahmed et al. (2006) [5] and Shibl et al. (2017) [13], which should be examined again to avoid the spread of contagious disease. This is especially important during the COVID-19 pandemic, as recent studies have noted [16,17,18,19,20]. The Center for Disease Control and Prevention (CDC) goes further in strongly recommending that hajjis receive a seasonal influenza vaccine, which is recommended for high risk pilgrims to reduce their own morbidity and mortality, but also to reduce transmission of the disease [3,18]. As raised by Barasheed et al. (2014) [14], Benkouiten et al. (2013) [15], and others more recently [16,17,18,19,20], the crowded conditions during Hajj and Umrah increase the risk of respiratory disease transmission including influenzas, tuberculosis, and Middle East Respiratory Syndrome (MERS) coronavirus, which was first identified in Saudi Arabia in 2012 [3,18,21,22]. In this study, 50.8% of pilgrims had reported that they were vaccinated against seasonal influenzas, which is higher than recently reported among Malaysian pilgrims (24.9%) [23], French pilgrims (31.8%), and Saudi Arabian, Qatari, and Australian pilgrims (31.0%) [14,15]. It should be considered whether the CDC recommendation should be added to the Umrah and Hajj visa requirements, although the evidence is still inconclusive and at times contradictory [16,17,18,19,20]. Whether greater enforcement of the regulations would help promote public health remains to be seen, but dramatically reducing the size of mass religious gatherings and adopting preventative measures has been shown to work at Hajj 2020 [19]. Following the COVID-19 pandemic these restrictions, and greater adoption of preventative measures such as the wearing of face masks may also be more acceptable to pilgrims.

Yet again, few pilgrims (*n* = 285, 28.1%) were vaccinated against poliomyelitis. The Saudi Arabian health authorities require proof of poliomyelitis vaccination from at least four weeks before arrival from pilgrims who are traveling from poliomyelitis reporting countries. This is an issue returned to again and again over the years by research teams including Memish et al. (2012a,2012b,2014, 2016, 2019), considering infectious disease prevention during Umrah and Hajj pilgrimages [8,22,24,25,26]. Of note, free polio vaccination is administered to pilgrims by Saudi Arabian MOH at the entry ports to promote the aim of world eradication of polio [7]. Perhaps there are more opportunities to offer further vaccinations at ports of entry.

The majority of respondents believed that the health education they had received was helpful (*n* = 661, 65.3%); however, some (*n* = 135, 13.3%) disagreed. Promoting health education and awareness programs prior to and during this spiritual journey is essential to improve their practices and to ensure lower risk of acquiring and spreading infections or existing medical conditions deteriorating, thus helping pilgrims to complete their rites safely and properly. The Saudi Arabian MOH has created an easy read guide to Hajj for Umrah pilgrims that comprises important information about vaccinations required and other preventative and precautionary measures [1,7]. These measures are founded in research following year on year Umrah pilgrimages [4,5,6,8,9,10,13,14,15,22,23,24,25,26,27,28].

Despite evidence to the contrary on information provided by the MOH in KSA [1,7], almost one quarter of the participants reported not receiving any health education and orientation around preventative measures (*n* = 213, 21.1%). This adds to the challenges in the management of the Hajj and Umrah and may have an impact on the services that are offered by the Saudi Hajj and Health authorities [7,10]. Clearly, despite all attempts to date to include advance provision of health education to pilgrims, there is more to be done in reaching the wider population. With hindsight, this study was an opportunity to establish which types of content and delivery of health education are seen as most helpful to pilgrims. This is an area for future research.

Of note, less than half of the pilgrims (*n* = 485, 47.9%) had reported using face masks “always or often”, and less than a third of participants (*n* = 305, 30.1%) had reported never using face masks in crowded areas; however, 63.2% reported lack of availability of these masks. This study showed an almost equal divide in those who do and do not use face masks in crowded places. This finding is in accordance with other studies in Saudi Arabia (56%) and the USA (42%) [19,24,25]. Since the outbreak of COVID-19 [27,28], the use of face masks has become universal in China and the rest of the world. However, the evidence of their protection against respiratory infections in the community is still controversial [16,17,18,20,28,29,30]. In Saudi Arabia, a study to estimate the incidence of Hajj-related acute respiratory tract infections (ARI) among pilgrims travelling from Riyadh revealed that the use of a facemask by men was a significant protective factor against ARI in Hajj, and fewer pilgrims (15.0%) had ARI compared with hajjis who used it sometimes (31.4%) or never (61.2%) [29].

It was reported in other studies that the use of face masks for more than eight hours can lead to a substantial decrease in the incidence of influenza-like illness (ILI) among pilgrims [14,29]. However, the effectiveness depended largely on adherence to mask use [30]. Hoang et al. (2019 and 2020) noted the complexity of the bacterial and viral RTIs [18] and high transmission rates despite preventative measures [19]. Adaptations to accommodation should be considered as the current use of large, single space dormitory style tents has been noted to encourage the spread of viral infections [16,18,19,20].

In the chain of transmission, regularly washing hands and the use of hand sanitizers resulted in significantly less ILI symptoms. A large proportion of the respondents (*n* = 840, 83%) always performed proper hand hygiene after coughing and sneezing. This percentage is higher than the French pilgrims in 2012 (46.3%) and USA pilgrims in 2009 (45.5%) [14,15]. Recent studies again emphasize the need for good hygiene practices and the ready provision of personal protective equipment [16,17,18,19,20].

In the present study, 852 pilgrims (84.1%) always used razor blades for shaving their heads, a practice which, if not done hygienically, poses a risk of transmission of blood-borne diseases such as HIV and Hepatitis B and C [5,18]. Some studies have identified issues with unlicensed barbers and sharing of razor blades amongst pilgrims [20].

The majority of pilgrims rated their experience with the Saudi Arabia Ministry of Health Services as “very satisfied” (*n* = 734, 72.9%). Of note is that health services including primary, secondary, and intensive care medical services are offered to millions of Umrah and Hajj pilgrims free of charge. Whether pilgrims felt open to express their opinions during completion of the survey is open to debate.

### Limitations of the Study

There are a number of biases that are present in this study, including the self-reporting and recall biases of participants. There may also be recruitment bias from the researchers at the airport in whom they approach and participation bias in those who decided to take part in this study and the responses (the number of refusals to participate was not recorded).

Although the sample size is reasonable, when compared with the cited literature, and considering the millions of Umrah pilgrims arriving at the holy cities in Saudi Arabia yearly, the findings may not be generalizable. Time constraints and language barriers were also limiting factors, since the recruitment was conducted at the airport.

The definition of health education was left open to the participant’s interpretation. This is an area that would benefit from closer attention in future studies to establish the content and type of information to include targeting the perhaps varying needs of different groups of pilgrims (language, literacy, economic aspects).

Despite these limitations, this study has currency and has provided several insights that were consistent with other studies, and some further insights on the behaviors and practices of pilgrims attending Hajj and Umrah in Saudi Arabia.

## 5. Conclusions

This study has identified several issues relating to the preventative measures that were practiced by Umrah pilgrims in Saudi Arabia in 1440H-2019. This highlighted inadequate meningitis and poliomyelitis vaccination and insufficient hygiene and safety practices, such as availability and proper wearing of face masks during Umrah. The Saudi MOH has the option to enforce mandatory proof of vaccination. The behavioral and practice issues amongst pilgrims have been shown in the literature to pose a high risk for acquiring respiratory and other infections. The worsening of pre-existing medical conditions is particularly relevant for Umrah pilgrims who tend to be older and with multi-morbidities. This is especially the case during the COVID-19 pandemic, given that these pilgrims fit the profile of those more likely to experience more severe symptoms and death. It is the duty and responsibility of each country in which their pilgrims participate in this holy rite to promote health education and raise awareness amongst individuals on basic preventative measures in order to promote public health. Further studies should focus on development of accessible health education content in a form that engages pilgrims from diverse backgrounds to promote comprehensive preventative measures during mass religious gatherings and pilgrimages.

## Figures and Tables

**Table 1 ijerph-18-00257-t001:** Responses to items on sociodemographic, level of education, and chronic diseases of participating pilgrims (*n* = 1012).

Sociodemographic Variables	Frequency	Percentage (%)
**Age groups**		
<20 year	51	5.0
20 ≤ 35 year	286	28.3
35 ≤ 50 year	342	33.8
50 ≤ 60	256	25.3
60 or more	77	7.6
**Gender**		
Male	656	64.8
Females	356	34.6
Prefer not to say		0.6
**Marital** **status**		
Widow	48	4.7
Single	172	17.0
Married	773	76.4
Divorced	19	1.9
**Level of education**		
Illiterate	58	5.7
Read/write	94	9.3
Primary/secondary	238	23.5
High school or above	622	61.5
**Existing chronic diseases**	230	22.7
**Type of chronic disease (*n* = 387) ***		
Diabetes mellitus	143	37
Cardiovascular diseases	164	42.4
Central nervous system disorders	4	1
Psychological/mental disorders	2	0.5
Immunocompromised travelers (sever-moderate)	5	1.3
Disabilities (visual, hearing, physical)	4	1
Others	65	16.8

* multiple answers were allowed.

**Table 2 ijerph-18-00257-t002:** Nationalities of respondent by frequency.

Country	*n*	%
Iraq	177	17.5
Egypt	161	15.9
Libya	160	15.8
Sudan	113	11.1
Indonesia	111	11.0
Pakistan	92	9.1
Afghanistan	80	7.9
Algeria	48	4.7
Tunisia	34	3.4
Jordan	30	3
Malaysia	22	2.2
Somalia	21	1.7
Kuwait	20	1.5
China	19	1.6
Philippines	19	1.9
Palestine	17	1.7
Tajikistan	15	1.2
United Kingdom	13	1.1
Morocco	11	1.1
United Arab Emirates	11	1.1
India	9	0.9
Turkey	9	0.9
Bangladesh	8	0.8
Niger	7	0.6
Nigeria	7	0.6
Burkina Faso	6	0.5
Chad	5	0.4
Germany	5	0.4
Holland	5	0.4
Oman	5	0.4
Senegal	5	0.4
Thailand	5	0.4
United States	5	0.4
Uzbekistan	3	0.3
Ethiopia	2	0.2
Syria	2	0.2
Yemeni	2	0.2
France	1	0.1
Kenya	1	0.1
Lebanon	1	0.1
Netherlands	1	0.1
Missing	27	2.4
Totals	1012	100%

**Table 3 ijerph-18-00257-t003:** Responses to items on aspects of vaccination history and the average duration of stay in Makkah and Madinah (*n* = 1012).

Vaccination History & Length of Stay	Frequency	Percent (%)
**History of vaccination (before travel) ***		
Not vaccinated	242	24
Influenza vaccine	514	50.8
Hepatitis A virus vaccine (HAV)	326	32.2
Hepatitis B virus vaccine (HBV)	310	30.6
Rabies vaccination	248	24.5
Meningitis vaccination	418	41.3
Poliomyelitis	285	28.1
Pertussis	254	25.1
**Duration of stay in Makkah**		
<1 week	218	21.5
1 week but <2 weeks	660	65.2
2 weeks–or more	134	13.2
**Duration of stay in Madinah**		
<1 week	614	60.6
1 week but <2 weeks	366	36.2
2 weeks or more	32	3.2

* multiple answers were allowed.

**Table 4 ijerph-18-00257-t004:** Responses to items on issues receiving health education (*n* = 1012).

Health Education	Frequency	Percent (%)
**Receiving health education**		
No	213	21.1
Yes	799	78.9
Self-education	185	23.2
**The site of receiving health education**		
In homeland	450	44.4
During arrival	201	19.8
At the residence inside Saudi Arabia	53	5.2
**Getting benefit from the received health education** (*n* = 1012)		
Yes	661	65.3
No	135	13.3
To some extent	216	21.3
**The sources of health education ***		
Press and publications	97	9.7
Family and friends	127	12.5
Lectures	262	25.8
Social media (Facebook, Instagram, Twitter, Snapchat, etc.)	51	5
Travel clinics	134	13.2
Health care providers	122	12
Saudi Ministry of Health website	16	1.6
Other websites	52	5.1
Others	182	18

* multiple responses permitted.

**Table 5 ijerph-18-00257-t005:** Responses to items on aspects of practicing of preventative measures by pilgrims (*n* = 1012).

Preventative Measures	Never *n* (%)	Rarely *n* (%)	Sometimes *n* (%)	Often *n* (%)	Always *n* (%)
Using umbrella or hat	509 (50.2)	75 (7.4)	130 (12.8)	154 (15.2)	144 (14.2)
Using sunblock creams	652 (64.4)	86 (8.5)	72 (7.1)	79 (7.8)	123 (12.2)
Drinking at least eight cups of water	13 (1.3)	37 (3.7)	151 (14.9)	173 (17.1)	638 (63.0)
Wearing face masks in crowded areas	305 (30.1)	74 (7.3)	148 (14.6)	134 (13.2)	351 (34.7)
Sufficient sleep	20 (2.0)	45 (4.5)	133 (13.1)	223 (22.0)	591 (58.4)
Avoid crowded areas	70 (6.9)	62 (6.1)	159 (15.7)	207 (20.5)	478 (47.2)
Using insect repellent spray	573 (56.6)	107 (10.6)	107 (10.6)	69 (6.8)	156 (15.4)
Use of personal tools (razor blades, etc.) without sharing with others	21 (2.1)	4 (0.4)	60 (5.9)	75 (7.4)	852 (84.2)
Dispose used tissue in waste bin	18 (1.8)	11 (1.1)	59 (5.8)	100 (9.9)	824 (81.4)
Washing hands with water, soap, or antiseptics					
- after coughing and sneezing	3 (0.3)	7 (0.7)	65 (6.4)	97 (9.6)	840 (83.0)
- before eating or preparing food	3 (0.3)	7 (0.7)	55 (5.4)	87 (8.6)	860 (85.0)
- after using the bathroom	3 (0.3)	5 (0.5)	47 (4.6)	90 (8.9)	867 (85.7)

**Table 6 ijerph-18-00257-t006:** Responses to items on aspects of availability and the sources of Personal Protective Equipment (PPE) (*n* = 1012).

PPE Availability	Brought It from Homeland *n* (%)	Was Given from Al Hamla *n* (%)	Not Available *n* (%)
Face masks	219 (21.4)	160 (15.8)	642 (63.4)
Personal tools	230 (22.7)	200 (19.6)	593 (58.0)
Sanitizers	240 (23.7)	162 (16.0)	612 (60.4)

**Table 7 ijerph-18-00257-t007:** Report the association between receiving some form of health education and practicing of preventative measures by pilgrims.

Association of Preventative Measures	No Median (Mean ± SD) Range	Yes Median (Mean ± SD) Range	(*p*)
Avoiding sun exposure using umbrella or hat	1 (1.9 ± 1.47)	2 (2.46 ± 1.6)	0.03 *
Using sun block creams	1 (1.45 ± 1.1)	1 (2.08 ± 1.51)	0.01 *
Drinking at least eight cups of water	4.4 ± 0.9	4.35 ± 0.95	0.59
Wearing masks in crowded areas	1 (2.56 ± 1.77)	4 (3.3 ± 1.60)	0.04 *
Sufficient sleep	4.74 ± 0.81	4.71 ± 0.7	0.85
Avoiding crowded areas	3.9 ± 1.4	3.97 ± 1.2	0.71
Using insect repellent	1 (2.02 ± 1.6)	1 (2.17 ± 1.5)	0.45
Healthy practice score	20.5 ± 4.9 7–35	22.7 ± 5.5 10–35	0.02 *

* *p* < 0.05 there is a statistically significant difference.

**Table 8 ijerph-18-00257-t008:** Responses to items on level of satisfaction with Saudi Arabia Ministry of Health Services (*n* = 1012).

Satisfaction with MOH Services	*n*	%
**The level of satisfaction**		
Very satisfied	734	72.5
Satisfied	212	21.0
To some what	54	5.3
Dissatisfied	9	0.9
Very dissatisfied	3	0.3

## Data Availability

The data presented in this study are available on request from the corresponding author. The data are not publicly available due to lack of a repository.

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
