# Peer review of "Assessment of Preventative Measures Practice among Umrah Pilgrims in Saudi Arabia, 1440H-2019"

_ijerph, 2020, doi:10.3390/ijerph18010257_

Round 1

Reviewer 1 Report

This study assesses preventative measures such as vaccination history and health education among Umrah pilgrims following their Umrah in Saudi Arabia. The assesses can provide insights which can then be used to slow down the spread of Covid-19 infection. While this research is important for current pandemic situations, some issues need to be addressed.

  1. One of the keywords is COVID-19, but COVID-19 is not mentioned in the abstract section at all.
  2. The research was conducted among pilgrims from February to April 2019. Wearing masks could be seasonal. It is helpful if the authors could provide data for other seasons, say during the winter period. The research was done before Covid-19 pandemic break and the behaviors are largely altered by the pandemic. So it would be more informative if there are data about wearing face masks during pandemic.
  3. Table 2, It would be better if the nations are listed according to the ratio number.
  4. Table 7, How is the association “between receiving of health education and practicing of preventative” calculated?
  5. Table 7, How to define “health education”? It is difficult to define the extent of receiving health education. Watching a TV show related to health and reading 10 professional medical books could be both regarded as receiving health education, but they are very different. So “receiving health education” could be very subjective. Is there a standard to measure it?

Author Response

Firstly, thank you for giving your time and expertise in reviewing this submission. It is greatly appreciated.

  1. Thank you, yes, this is now included in the Abstract.

  1. You point out an area which needs more research which we have now added to the Conclusions. We were unable to find seasonal reports for Umrah but have added these recently published articles, covering different times of year, to both the Introduction and Discussion sections:

Alasmari AK, Edwards PJ, Assiri AM, Behrens RH, Bustinduy AL (2020) Use of face masks and other personal preventive measures by Hajj pilgrims and their impact on health problems during the Hajj, Journal of Travel Medicine, taaa155, https://doi.org/10.1093/jtm/taaa155

Alfelali M, Haworth EA, Barasheed O, Badahdah A-M, Bokhary H, Tashani M, Azeem MI, Kok J, Taylor J, Barnes EH, El Bashir H,Khandaker G, Holmes EC, Dwyer DE, Heron LG, Wilson GJ, Booy R, Rashid H (2020) Facemask against viral respiratory infections among Hajj pilgrims: A challenging cluster-randomized trial. PLoS ONE 15(10): e0240287. https://doi.org/10.1371/journal.pone.0240287

Hoang VT, Gautret P, Memish ZA & Al-Tawfiq JA (2020) Hajj and Umrah Mass Gatherings and COVID-19 Infection. Curr Trop Med Rep 7, 133–140 (2020). https://doi.org/10.1007/s40475-020-00218-x

Hoang V-T, Dao T-L, Ly TDA, Belhouchat K, Chaht KL, Gaudart J, Mrenda BM, Drali T, Yezli   S, Alotaibi B, Fournier PE, Raoult  D, Parola P, De Santi V & Gautretet P (2019) The dynamics and interactions of respiratory pathogen carriage among French pilgrims during the 2018 Hajj. Emerg Microbes Infect. 2019;8:1701–10.

Liang M, Gao L, Cheng C, Zhou Q, Uy JP, Heiner K, Sun C (2020) Efficacy of face mask in preventing respiratory virus transmission: A systematic review and meta-analysis. Travel Med Infect Dis. 2020 Jul-Aug;36:101751. doi: 10.1016/j.tmaid.2020.101751. Epub 2020 May 28. PMID: 32473312; PMCID: PMC7253999.

  1. Table 2 has been re-ordered.

  1. & 5 . Again, you raise very valid points. This was a question asked of pilgrims so only represents their views on whether they self-report as having received any form of health education (Q10 yes/no) and whether they adopt preventative measures (Q14). This has been raised as a limitation in the manuscript and added to the conclusions as an area for further research.

Many thanks for sharing your insightful and constructive comments.

Reviewer 2 Report

  1. Perhaps a slightly more elaborated introduction which may point out the significance of this study has been discussed in the introduction section. Offering a more rich context of why assessment of preventative measures practice among Umrah Pilgrims in Saudi Arabia is critical. 
  2. Need to highlight research gaps and provide evidence on the need of the study. It is also suggested to provide evidence on the research studies alike in other preventative measures practice contexts.
  3. The discussion in this study should be based on the statistical analysis result, and constructive implications should be highlighted. The statistical analysis result and the interpretation of the researcher should be filled with specific constructive details that can be applied to the actual field of assessment of preventative measures practice among Umrah Pilgrims in Saudi Arabia; however, this part is insufficient in this study. Therefore, the implications should be found and highlighted, so that the study results could be applied at the actual field of preventative measures practices.
  4. The discussion section should be extended by a clear focus of how the research problem can be solved and how the research questions (RQ) can be answered on the base of the empirical insights.
  5. The article is very difficult to read. In part, it is because of its topic and various analyses; however, the way in which the authors explain their reasoning is not straightforward.

Author Response

Thank you for giving your time and expertise in reviewing this submission. It is greatly appreciated and the paper will be stronger for incorporating your suggestions and making those amendments.

  1. Five recent articles have been added to the Introduction and Discussion sections (please see response to Reviewer#1 above in item 2).
  2. Thank you, yes, we have elaborated on the 5 additional articles mentioned above and made what we hope you agree is a clearer statement justifying the study at the end of the Introduction..
  3. Thank you. We have revised and expanded the Discussion section with the additional references noted above. Insights have been highlighted at the end f each paragraph in the Discussion section to extrapolate outcomes.
  4. Thanks again. The Conclusion has been amended and extended to restate the problem and clarify the options in light of the study outcomes.

Thank you for raising this issue. We hope you agree that the language has been simplified.

Round 2

Reviewer 1 Report

I have carefully read the revised manuscript and point-for-point, and the authors have adequately addressed my major concerns. They now provide additional data and have added further discussions. The efforts in revising the manuscript have largely strengthened the manuscript.

Author Response

We are very much thankful to the reviewer for the deep and thorough review and the positive comments. Thank you again for the useful suggestions and comments.

Kind regards,

Reviewer 2 Report

  1. As before, I appreciate the authors' responses to my initial round of comments. In my view, the manuscript is much improved, and the conceptual layout of the front-end now aligns more closely with the methodology and subsequent interpretation.
  2. Lastly, the study needs more cohesion in presenting ideas and significant findings, which also should clearly emphasize which findings or method usage of the research was new and original. Also the theoretical contribution to offer deeper insight into the literature should be more clarified.

Author Response

Thank you for giving your time and expertise in reviewing this submission.

Authors are grateful to the reviewer for the positive and encouraging comments. Considering the reviewer useful suggestion, the article has been amended.

Thank you again,